# Genome-Wide Genetic Structure of Henan Indigenous Chicken Breeds

**DOI:** 10.3390/ani13040753

**Published:** 2023-02-19

**Authors:** Yihao Zhi, Dandan Wang, Ke Zhang, Yangyang Wang, Wanzhuo Geng, Botong Chen, Hong Li, Zhuanjian Li, Yadong Tian, Xiangtao Kang, Xiaojun Liu

**Affiliations:** 1College of Animal Science and Technologyw, Henan Agricultural University, Zhengzhou 450046, China; 2Henan Key Laboratory for Innovation and Utilization of Chicken Germplasm Resources, Zhengzhou 450046, China; 3International Joint Research Laboratory for Poultry Breeding of Henan, Zhengzhou 450046, China

**Keywords:** Henan indigenous chicken, breeding, genetic diversity, gamecock, selective sweep

## Abstract

**Simple Summary:**

There are five indigenous chicken breeds in Henan Province, China; however, comprehensive knowledge of their genetic basis is lacking. Therefore, using whole genome resequencing, we examined the genetic make-up, genomic diversity, and migration history of the indigenous Henan chicken populations as well as the selection factors and genes behind the distinct phenotypes of Henan gamecocks. These results will make it easier to comprehend the traits of the germplasm and the potential for using native breeds from Henan.

**Abstract:**

There are five indigenous chicken breeds in Henan Province, China. These breeds have their own unique phenotypic characteristics in terms of morphology, behavior, skin and feather color, and productive performance, but their genetic basis is not well understood. Therefore, we analyzed the genetic structure, genomic diversity, and migration history of Henan indigenous chicken populations and the selection signals and genes responsible for Henan gamecock unique phenotypes using whole genome resequencing. The results indicate that Henan native chickens clustered most closely with the chicken populations in neighboring provinces. Compared to other breeds, Henan gamecock’s inbreeding and selection intensity were more stringent. TreeMix analysis revealed the gene flow from southern chicken breeds into the Zhengyang sanhuang chicken and from the Xichuan black-bone chicken into the Gushi chicken. Selective sweep analysis identified several genes and biological processes/pathways that were related to body size, head control, muscle development, reproduction, and aggression control. Additionally, we confirmed the association between genotypes of SNPs in the strong selective gene *LCORL* and body size and muscle development in the Gushi-Anka F2 resource population. These findings made it easier to understand the traits of the germplasm and the potential for using the Henan indigenous chicken.

## 1. Introduction

Indigenous chicken breeds are gradually developed under the influences of long-term natural environmental and artificial selection, and display abundant phenotypic diversities, such as heterogeneous feather colors, comb, and plumage shapes [1], which are unique resources for genetic improvements in chickens. Surveying for genetic variation can contribute to further research on the molecular mechanisms of chickens’ evolution and domestication, and the findings of such a survey can serve as a valuable reference for improving the breeding and cultivation of Chinese indigenous breeds. There are five indigenous chicken breeds in Henan Province, China, including the Gushi chicken (GS), Zhengyang sanhuang chicken (ZYSH), Xichuan black-bone chicken (XCBB), Lushi blue-shelled-egg chicken (LS), and Henan gamecock (HNG). They have unique phenotypic traits that distinguish them in terms of morphology, behavior, skin and feather color, and productive abilities. However, the characteristics of genetic variations in the breeds are still unclear. Theoretically, a thorough understanding of the genetic diversity of the indigenous breeds might support conservation and utilization through suitable breeding programs in the future.

In many domestic animals, whole genome resequencing technology has already been used to examine the genetic composition of populations [2,3,4,5]. Based on this technology, Luo et al. (2020) investigated the genetic diversity of 157 native Chinese chickens [6], and discovered that isoprenoid synthase domain containing (*ISPD*) may be mandatory for the muscularity of gamecocks, whereas alkylglycerol monooxygenase (*AGMO*) and carboxypeptidase Z (*CPZ*) are crucial for determining the behavioral features. Wang et al. (2020) analyzed the origin and domestication of the chicken using 863 birds worldwide and identified that Southwestern China, Northern Thailand, and Myanmar were the main domestication centers of domestic chickens [5].

Long-term selection alters the patterns of variation in certain genomic regions leading to increased allele frequencies, the degree of linkage disequilibrium, and a reduction local diversity [7,8]. Simultaneously, the genomic regions linked to the sites are fixed due to the genetic hitchhiking effect [9]. The distinctive genetic traces or imprints left in the genomic regions are signatures that underwent selection [10], which can be effectively detected by selective sweep analysis [11,12,13]. Different statistical parameters, such as fixation index (FST), nucleotide diversity (π), and reduction of diversity (ROD), have been proposed according to the investigated selection signatures that are within populations or between populations [14,15]. Among these methods, the FST is usually used to quantify the degree of population genetic differentiation between populations [16]. The π represents nucleotide polymorphisms, and decreases with increasing levels of selection. The detection power of selection signatures has been suggested to be improved by combining several statistical methods [17]. Numerous chicken selection signatures have been studied in depth, including those related to feather color [18], comb size [19], and skin color [20], as well as those relating to bidirectional selection of body weight [21], and regional and commercial breed features [22].

In the present study, 50 Henan indigenous chickens were whole genome resequenced. Combined with the genome sequencing data of another 11 Chinese nationwide canonical indigenous chicken breeds, we comprehensively investigated the genetic structure of the five Henan indigenous chicken breeds and identified the unique selective genomic variants/genes in HNG.

## 2. Materials and Methods

### 2.1. Sample Collection and Sequencing

The 50 blood samples from the 5 Henan indigenous chicken breeds, including 10 HNG, 10 GS, 10 LS, 10 XCBB, and 10 ZYSH, were collected. Genomic DNA was extracted from blood samples using the TianGen DNA Kit (DP 341, Tiangen Biochemical Technology, Beijing, China). Using a NanoDrop spectrophotometer 2000 (NanoDrop Inc., Wilmington, DE, USA) and agarose gel electrophoresis, the quantity and quality of the genome’s DNA were assessed. The 50 samples were resequenced on whole genome level by using the DNBSEQ-T7 platform at the Wuhan Benagen Technology Co., Ltd. (Wuhan, China).

The WGSs of 115 chickens from another 11 chicken breeds, including 10 Guangxi Yao chicken (YAO), 10 Hetian chicken (HT), 10 Huaixiang chicken, 10 Huanglang chicken (HL), 10 Huiyang bearded chicken (HUX), 10 Jianghan chicken (JH), 10 Ningdu Yellow chicken (ND), 10 Wenchang chicken (WC), 10 Wuhua Yellow chicken (WH), 10 Huaibei partridge chicken (HBM), and 15 Red jungle fowl (RJFt) were retrieved from a published dataset [22,23].

### 2.2. Whole Genome Variants’ Identification and Annotation

To create clean reads of premium quality, the stringent quality filtering techniques listed below were used: (1) deleting reads with ≥10% undefined nucleotides (N); (2) eliminating reads with more than 50% bases with phred scores of ≤20; and (3) disposing reads aligned to the barcode adaptor. The clean reads were then aligned using the BWA-MEN alignment function built into the BWA (v0.7.17, Heng Li, USA) [24] to the chicken reference genome (GRCg6a). Duplicated reads were removed using the Picard package [25]. To produce the sequencing coverage statistics, the bedtools (v2.29.2, Aaron R. Quinlan, USA) was used [26]. Using the GATK (v3.8, Aaron McKenna, USA) software, variant calling was carried out [27]. Single nucleotide polymorphisms (SNPs) were filtered by the GATK’s Variant Filtration with options “QD < 2.0 || MQ < 40.0 || FS > 60.0 || SOR > 3.0 || MQRankSum < −12.5”, excluding those exhibiting segregation distortions or sequencing errors. Finally, a total of 9,861,819 SNPs with minor allele frequency >0.05 and maximum miss rate >0.8 were obtained using VCFtools (v0.1.16, Adam Auton, UK) [28] for subsequent analysis. The Ensemble genome database and SNPEff (v 4.1, Pablo Cingolani, USA) [29] program was used to obtain information about SNPs annotation.

### 2.3. Heterozygosity and Runs of Homozygosity

To characterize genetic diversity, we employed a variety of metrics that we all acquired through PLINK (v1.90, Shaun Purcell, USA) [30].

VCFtools v0.1.16 [28] was used to estimate the nucleotide diversity across the entire genome for each population. The ratio of observed heterozygosity to observed homozygosity (Ho/-het) was calculated as 1–(number of observed homozygous loci divided by the number of non-missing loci), and the expected heterozygosity (He) was estimated as the 1–(number of expected homozygous loci over number of non-missing loci). All SNPs were averaged to determine the observed heterozygosity and expected heterozygosity estimates for each population’s members. Based on the Runs of Homozygosity (ROH), the inbreeding coefficient of each individual was estimated. Using PLINK (v1.90, Shaun Purcell, USA), long homozygous fragments were scanned. The following specific parameters were used to assess homozygosity: a sliding window of 50 SNPs along the chromosome was used; each sliding window allowed a maximum of one heterozygote, five missing SNPs, a minimum length of ROH of 100 kb, a minimum density of one SNP per 50 kb, and a maximum interval between consecutive SNPs of 1000 kb was permitted for each sliding window.

According to McQuillan [31], the inbreeding coefficients (F_ROH_) for each breed were determined using the following formula:
$$F_{ROH}=\sum^{\text{​}}\frac{L_{ROH}}{L_{AUTO}}$$


L_AUTO_ is the length of the autosomal genome that spans the SNP locations (960280 kb in the present study).

### 2.4. Analysis of Population Structure, Linkage Disequilibrium, and Gene Flow

Neighbor in the PHYLIP (v3.69, Joseph Felsenstein, USA) package was used to create neighbor-joining (NJ) relationship trees between participants [32], which were then presented by MEGA (v7.0, Sudhir Kumar, USA) [33]. PLINK (v1.90, Shaun Purcell, USA) was used to perform principle component analysis (PCA) on the genetic distance matrix of 16 breeds. To avoid artifacts due to linkage disequilibrium (LD) and to save computation time, SNPs with high-wise R2 values (R > 0.2) were pruned from the dataset using PLINK (v1.90, Shaun Purcell, USA, arguments: --indep-pairwise 50 5 0.2), and SNPs having high-wise R2 values (R > 0.2) were removed from database [30]. The supervised ADMIXTURE (v1.3.0, David Alexander, USA) program [34] was used to study the population structure through the maximum likelihood model. The number of genetic clusters, K, was predetermined and ranged from 2 to 16. Using PopLDdecay [35], LD for 16 breeds was determined based on the correlation coefficient R2 statistics of the two loci. For the purpose of analyzing gene flow among the 16 breeds, a population phylogenetic tree was built using TreeMix (v1.13, Robert R Fitak, USA) [36].

### 2.5. Selective Sweep Analysis

Fst and θπ have been shown to be successful for discovering regions of selective elimination, particularly when mining functional zones strongly associated to the living environment, where strong selection signals can be acquired. We used the sliding-window approach (40-kb windows with 10-kb increments) to determine the genome-wide distribution of FST values and θπ ratios among HNG as well as non-Henan game chickens (NHNG) in order to identify possible areas that had undergone directional selection in the HNG. Based on the ratio of π for a subpopulation in relation to a control subpopulation, the reduction of diversity (ROD) values were estimated. FST values were changed using Z-transform, and θπ ratios were transformed using log_2_ ratio. The process was analyzed using vcftools [28]. FST and log_2_(θπ ratio) joint analysis, including HNG and NHNG, revealed the genomic region imprint of HNG. We looked at the windows with the top 5% values for both the FST and log_2_ ratio as potential candidates for strongly selected genes. Finally, the candidate genes were analyzed by Gene Ontology categories (GO) and Kyoto Encyclopedia of Genes using Metascape https://metascape.org/gp/index.html#/main/step1 (accessed on 11 November 2022).

### 2.6. Association Study between SNPs in the LCORL Gene and Growth and Carcass Traits

The *LCORL* gene SNPs were identified by using double-digest genotyping-by-sequencing (ddGBS) data from the population of 734 Gushi × Anka F2 chickens [37]. Using Haploview, linkage disequilibrium analysis of SNPs was carried out. The genotyping information for SNPs was utilized to conduct an association study with growth and carcass traits.

### 2.7. Statistical Analysis

The generalized linear mixed model (GLM) included with SPSS 23.0 (IBM, Chicago, IL, USA) was used to determine the relationship between SNPs and growth, carcass, and meat quality variables. The models used were as follows:Yiklm = μ+ Gi + Hk + fl + eiklm(1)
Yiklm = μ+ Gi + Hk + fl + b (Wiklm - ) + eiklm(2)

Model I was employed to analyze SNP association with meat quality and growth variables, whereas Model II, which included carcass weight as a covariate, was employed to investigate SNP relationship to carcass traits. These models used Yiklm as the dependent variable (phenotypic value), μ as the observation mean, eiklm as random error, Gi as the genotype fixed effect (i = genotypes), Hk as the hatching fixed effect (k = 1, 2), fl as the family fixed effect (l = 1, 7), b as the carcass weight regression coefficient, and Wiklm as the slaughter weight of the individual, which was the average slaughter weight [37]. Least significant difference (LSD) was used to determine the statistical differences among the least squares means of the different genotypes.

## 3. Results

### 3.1. Genomic Variants in the Henan Indigenous Chicken

A total of 50 individuals from five Henan indigenous chicken breeds were whole genome resequenced with an average sequencing depth 18-fold of the genome. Approximately 25.2 million autosomal SNPs were reported. Following filtering of minor allelic frequencies <0.01 and call rates <0.9, around 9.86 million autosomal SNPs were eventually kept and utilized in subsequent experiments. Most of the variations were annotated in introns (57.812%), exons (1.853%), intergenic region (19.667%), upstream (10.141%), and downstream (9.625%) of genes (Appendix A). Chromosome SNP density distribution analysis indicated that the SNPs were evenly distributed on each chromosome, other than at the telomeres of some chromosomes, and chromosome 1 and 2 had the highest densities of SNPs (Figure. S1).

### 3.2. Population Genetic Structure and Genetic Diversity

In order to explore the population genetic structure of Henan indigenous chickens and its relationship with other chicken breeds in other parts of China, the neighbor-joining (NJ) tree based on genome-wide SNPs of 16 chicken breeds was constructed, and the results showed that all the chickens could be divided into three large clusters (Figure 1A). Of them, cluster 1 included the RJFt and WC, cluster 2 included HUX, WH, YAO, and HX, and cluster 3 included the remaining 10 indigenous chickens, including five Henan indigenous chicken breeds, HBM, JH, HL, ND and HT. Obviously, a close genetic relationship between GS and HBM was found. In addition, the HNG showed conspicuous separation from other Henan indigenous chicken breeds.

Further principal component analysis (PCA) divided the 16 chicken populations into five groups (Figure 1B). Group 1 was the RJFt population, and the individuals in the population were scattered, which reflected the large disturbance in the genome variation of the population. Group 2 contained five chicken breeds (HUX, WH, YAO, HX, and WC) located in Southern China. Group 3, 4, and 5 were the three subclasses of cluster 3 of the NJ tree.

Additionally, we carried out an unsupervised admixture analysis, with K ranging from 2 to 16, to estimate the degree of admixture among 16 breeds of chicken (Appendix A). When K = 2 as the lowest cross-validation error, there was a clear genetic divergence between the RJFt population and other populations. When K = 4, the genetic divergence was consistent with the aforementioned PCA result (Figure 1B). The RJFt population and HNG population were independently separated from the other populations. Meanwhile, Henan indigenous chickens, except for HNG, formed a group with HBM, and WC, WH, YAO, HX, and HUX formed another group, while HT, HL, ND, and JH populations had obvious widespread genetic introgression from other populations. At K = 5, LS and XCBB populations were separated from Henan indigenous chickens (Figure 1C). As a whole, the HNG population had the purest genetic background, while the other four populations had slight genetic introgression from chicken populations in other regions of China, such as HUX, HX, WC, WH, and YAO. In addition, the genome of the HNG population was also slightly infiltrated into the other four Henan indigenous chicken breeds (Figure 1C).

The genetic diversity (heterozygosity and nucleotide diversity) of all the 16 chicken breeds was also estimated. The results showed that the expected heterozygosity (He) and nucleotide diversity (π) of HNG population were the lowest, followed by the RJFt populations and other indigenous chicken populations (Table 1). For Henan indigenous chickens, with the exception of HNG, their SNP diversities and heterozygosity were at a medium-to-high level compared with other local chicken breeds. In addition, the lower He and π of RJFt population implied that the RJFt genetic drift was caused by long-term small population rearing in different regions.

### 3.3. Runs of Homozygosity and Linkage Disequilibrium Unveiled Genome-Wide Genetic Variation Remodeling of Henan Gamecock by Strong Artificial Selection

Runs of homozygosity can reveal an animal’s degree of inbreeding. While a short run of homozygosity indicates more distant shared ancestors, a long run of homozygosity indicates inbred animals with recent common ancestors. We compiled the genomic inbreeding coefficient (F_ROH_) and runs of homozygosity for 16 different chicken breeds (Figure 2A–C). The results showed that, with the exception of the RJFt, Henan indigenous chicken populations had longer runs of homozygosity and higher F_ROH_, suggesting that Henan indigenous chickens had a higher inbreeding degree among all the 15 Chinese indigenous chicken breeds. Furthermore, of the Henan indigenous chickens, HNG had the largest value and the longest runs of homozygosity, and the highest F_ROH_, indicating that HNG was subject to more strict inbreeding.

Linkage disequilibrium (LD) decay patterns can provide details about a population’s evolution. As a result, comparing the rate of LD decay between populations might provide useful information about the overall variety of a species. The results showed that the decay rate of Henan indigenous chickens was slower, indicating that these populations had experienced stronger selection comparing to other populations. Not surprisingly, HNG chickens showed the slowest rate of LD decay, suggesting that the breed could have been subjected to intense artificial selection during domestication (Figure 2D).

### 3.4. TreeMix Analysis Revealed the Migration History of Henan Indigenous Chickens

Given that a potential introgression from chicken breeds in other regions of China to Henan indigenous chicken breeds has been suggested by above Admixture analysis, to better understand the migration patterns of Henan indigenous chickens, we further reconstructed a maximum likelihood (ML) tree by TreeMix to examine populations split and migration events. In this ML tree (Figure 3), we observed an early split between central (GS, HNG, ZYSH, XCBB, LS, HBM, and JH) and southern (HUX, WC, HX, YAO, WH, HT, ND, and HL) populations. Gene flows from southern chicken breeds into ZYSH and from XCBB into GS could be evidenced, which conformed with the above admixture results.

### 3.5. Genome-Wide Selective Sweep Signals in Henan Gamecock

We found from the ROH and LD decay that Henan fighting cocks were strongly selected and had distinctive personality traits. In order to further investigate and explore the genes associated with selection in fighting chickens, we conducted a genome-wide selection signal analysis in HNG. After filtering, 9,861,819 SNPs in HNG were retained for the analysis of selection signals between gamecocks and non-gamecocks. Based on both the ROD and FST statistical methods, the top 5% selected genome regions were considered as a potential selection signal region, and a total of 1103 candidate genes were identified from FST, while 1182 genes were obtained from ROD (Figure 4A,B, Appendix A). A total of 399 overlapping selective genes were obtained in HNGs after gene intersection obtained by the two statistical methods (Figure 4C, Appendix A). Functional enrichment analysis indicated that the 399 candidate selective genes were mainly involved in mesenchyme development, cell adhesion molecules, muscle structure development, cell morphogenesis, MAPK cascade, and regulation of chondrocyte differentiation modulation of chemical synaptic transmission (Figure 5A,B).

The genomic regions with the most significant selective signals occurred in chromosome 1: 42,880,001–42,970,000, containing the transmembrane and tetratricopeptide repeat containing 3 (*TMTC3*) gene, and chromosome 4: 75,820,001–75,870,000, containing the ligand dependent nuclear receptor corepressor like (*LCORL*) gene and non-SMC condensin I complex subunit G (*NCAPG*) gene. The shared long-range haplotypes across the HNG population could be observed in *TMTC3*, and *LCORL*–*NCAPG* genes, respectively (Figure 6A,B). Furthermore, we analyzed the association between genotypes of SNPs in the LCORL gene and growth and carcass traits in the Gushi-Anka F2 resource population. Based on the previous genotyping-by-sequencing (GBS) data of the population [37], we detected six SNPs on the *LCORL* gene (Appendix A), two (Chr4: 75,854,181C>T and Chr4: 75,859,000G>A) of which were significantly associated with body weight (BW) at 0, 2, 4, 6, and 12 weeks of age; body size index including chest depth, breast bone length, body slanting length, and pelvis breadth; and carcass traits including semi-evisceration weight, eviscerated weight, semi-evisceration weight rate, eviscerated weight rate, breast muscle weight, leg muscle weight rate, and the other ratio of viscera (Appendix A). For the SNP Chr4: 75,859,000G>A, the genotype AA was conducive to weight gain and chicken meat production (Figure 7A). The allele frequency of A was higher than that of G in broiler breed (Cobb), while reduced in gamecock breeds, and was lowest in other indigenous and layer breeds (Figure 7B). These results further supported that the strong selection of *LCORL* gene in HNG was related to body size and muscle development.

### 3.6. Common and Unique Selection Characteristics of Henan Gamecock

To further identify the selected genomic regions and genes that were unique to HNG and common to other gamecock chicken breeds, we integrated the candidate selective genomic regions of HNG with the candidate genomic regions of Chinese gamecocks, which were previously reported by Luo et al. (Appendix A). Ref. [6], and identified 52 genes common to gamecocks (Appendix A) and 347 genes unique to HNG (Appendix A). These common selective genes were mainly involved in cell morphogenesis, neuron regulation, behavior, muscle organ development, female gonad development, regulation of secretion by cell, and positive regulation of glucose transmembrane transport (Figure 8A), while the unique selective genes in HNG were mainly involved in sensory organ development, neuron projection development, DNA metabolic process, epithelial cell differentiation involved in kidney development, protein ubiquitination, regulation of the Wnt signaling pathway, manipulation of chondrocyte differentiation, and muscle structure development (Figure 8B).

## 4. Discussion

Indigenous chicken breeds usually have excellent meat and egg quality, strong stress resistance, and possess plentiful genetic diversity and a strong ability to adapt to the environment [38]. These indigenous varieties are valuable genetic resources for economic traits improvement and breeding of new varieties. The protection and utilization of the genetic resources are of great significance for the sustainable development of the poultry industry.

Genome resequencing technology, widely used in study of the genetic characteristics and population admixtures [39,40,41,42], has accelerated the progress in resolving the genetic roots of numerous complicated phenotypic features including chicken body size [43] and plumage color in ducks [44]. Here, we, for the first time, reported the genetic structures and the migration history of the five Henan indigenous chicken breeds, and the unique genomic characteristics of HNG by whole-genome resequencing analysis.

Comparative genomics analysis was performed based on genome-wide SNPs of the Henan indigenous chicken breeds, other indigenous chickens, and wild ancestors. Population structure analysis revealed the overall differences and similarities of the genomic architecture of the five Henan indigenous chicken breeds and other Chinese indigenous chicken breeds as well. The NJ-tree and PCA analysis indicated that the clustering patterns of the chicken population were closely related to geographical location and altitude, which is consistent with the clustering patterns of yellow feather chickens in China [18]. For example, the Henan indigenous chickens had the closest clustering with the chicken populations in neighboring provinces (JH in Hubei province and HBM in Anhui province), but gathered in different clusters with the chicken populations in the Southern Guangdong province (WH, HX) and Hainan province (WC). In addition, it was worth noting that the HNG was obviously separated from the other Henan indigenous chickens (PCA), and was not genetically infiltrated by other non-gamecock breeds, maintaining a relatively pure genetic background (admixture), which might be due to the fact that the breeding of gamecock chickens avoids crossbreeding with other types chickens in order to prevent the reduction of aggression [45].

Admixture analysis indicated that, compared with other chicken breeds in Henan Province, LS and XCBB populations have a closer genetic relationship (K = 5). According to the records of annals of poultry genetic resources in China, both LS and XCBB chickens have black legs and lay blue-shelled eggs, though there are obvious differences in feather, skin, and meat colors. These may be explained by the closer genetic relationship between the two breeds. More interestingly, we found that GS and HBM remained unseparated until K = 15, indicating that the two breeds probably originated from a recent common ancestor, which can also be explained by almost identical phenotypes and very close geographical proximity. HNG evolved from the red jungle fowl according to an analysis of migratory patterns among different types. Another gene flow was seen between southern chicken breeds and ZYSH. The chicken breeds of southern origin used in the study mostly exhibit three yellow characteristics, namely yellow feathers, beaks, and leg skin, which are similar to the characteristics of ZYSH. Since ancient times, the Henan area has been a center of trade and information exchange, and it is clear that trade, migration, and cultural exchange between people from the two regions facilitated the spread of chicken variants [46].

The evaluation of genetic diversity showed that the heterozygosity of Henan indigenous chickens, except HNG, were higher, indicating that the diversity protection of these local breeds was highly valued. The amount of animal inbreeding and the population history are reflected in runs of homozygosity in the genome. Cattle have already shown that runs of homozygosity are useful for estimating animal inbreeding [3,47], as well as in goats [48] and pigs [49]. In the Chinese indigenous chickens included in our study, the length of ROH and F_ROH_ varied greatly among different breeds. Except for the RJFt, HNG had the longest ROH and the highest F_ROH_, which were consistent with the findings by Zhang et al. [45], in which the game chickens have stronger inbreeding than other local breeds. In addition, previous research had confirmed that LD increased with the inbreeding rate and decreased with an increase in hybridization [50]. The obvious increase of LD in HNG further confirmed strong artificial selection had subjected the gamecocks to more strict inbreeding to maintain breed-specific traits.

Furthermore, we identified the selective genomic regions and genes that were unique to HNG and common to other gamecock chicken breeds based on genome-wide selective sweep signal analysis. However, only 52 common selective genes were identified from the 399 candidate selective genes in HNG, similar to the previous selection of Xishuangbanna gamecock [51], which may be due to genetic drift or artificial selection differences between different breeds [6]. Among the common selective genes, some have been clearly reported to be associated with body size, insulin like growth factor 2 mRNA binding protein 1 (*IGF2BP1*), insulin like growth factor 1 (*IGF-1*), *LCORL*, calmodulin-lysine N-methyltransferase (*CAMKMT*) [52,53], head size alkylglycerol monooxygenase (*AGMO*), CASP2 and RIPK1 domain containing adaptor with death domain (*CRADD*) [54], pea-comb SRY-box 5 (*SOX5*) [55], muscle development collagen type VI alpha 1 chain (*COL6A1*), isoprenoid synthase domain containing (*ISPD*) [56,57], egg production and ovarian development estrogen receptor 1 (*ESR1*), prolyl endopeptidase-like (*PREPL*) [58] and neuropsychiatric disorders karyopherin subunit alpha 3 (*KPNA3*), hepatocyte growth factor (*HGF*), ectodysplasin A (*EDA*), cadherin 8 (*CDH8*), AKT interacting protein (*AKTIP*) [59,60,61,62,63]. In addition, the common selective genes including CDH8, ephrin B1(*EFNB1*), ETS variant 1(*ETV1*), hepatocyte growth factor (*HGF*), slit guidance ligand 2(*SLIT2*), cyclase associated actin cytoskeleton regulatory protein 1 (*CAP1*), *IGF2BP1*, major facilitator superfamily domain containing 2A (*MFSD2A*), and *ISPD*, were most significantly enriched in cell morphogenesis, such as cell/neuron projection morphogenesis and cell morphogenesis involved in differentiation, which played a crucial role in shaping the physiological functions of a given organ, such as cell shape change [64]. Importantly, cell shape changes are important drivers of growth, muscle strength, and neurotransmitter transmission [65,66]. These selective genes may be conducive to the body size and muscle strength control of gamecock chickens and the neuroregulation of their aggressive behavior. In addiiton, several biological processes involved in muscle development, regulation of temperature, and neuron synapse structure were indeed enriched. Among the unique selective genes in HNG, such as mono-amine oxidase A (*MAOA*) [67,68], glutamate metabotropic receptor 8 (*GRM8*) [69], and orthologs from chicken RNA binding protein, fox-1 homolog 1 (*RBFOX1*) [70] were involved in impulsive aggressive behavior. The candidate gene suppressor of cytokine signaling 2 (*SOCS2*) potentially affected immune control and body size [71]. The candidate gene myosin, heavy chain 1E (*MYH1E),* was involved in the muscular structure and processes related to muscle fiber regeneration and repair [72]. Nudix hydrolase 7 (*NUDT7*) was the most likely candidate gene responsible for the redness of meat color in pork [73]. These genes uniquely affected phenotypic selection characteristics of HNG.

Among these selective genes, *LCORL* was located in the most strongly selected genomic region in HNG. Genomic variations in the *LCORL* gene had been reported to be associated with the body size of horses [74,75,76], cattle [77], sheep [45], and donkeys [78]. In addition, in the F2-generation population derived from the intercrossing of the Luxi gamecock and the white feather broiler chicken, *LCORL* was significantly associated with full eviscerated weight [79]. In our study, genotypes of SNPs Chr4: 75859000G>A and Chr4: 75859000G>A in the *LCORL* gene were also identified to have an obvious association with the early body weight and body size at different weeks of age, and carcass traits in the Gushi-Anka F2 chicken population. By rationally combining the aforementioned studies, we may suggest that variations in *LCORL* may be crucial in determining the muscular characteristics of gamecock hens.

## 5. Conclusions

In summary, we comprehensively characterized the population genetic structure, genome diversity, and migration history across all five Henan indigenous chickens, and revealed that Henan indigenous chickens, except HNG, had a relatively high genetic diversity, while HNG had been subjected to more strict inbreeding to maintain breed specific traits. In addition, a slight genetic introgression from chicken populations in other regions of China into Henan indigenous chickens, except HNG, occurred. Moreover, we identified several common selective genes and biological processes/pathways in gamecocks that were related to body size, head control, muscle development, reproduction, and aggression control. Importantly, we detected the unique selective genes and biological processes related to sensory organ development and regulation of chondrocyte differentiation in HNG, which reflected the rapid response to external stimuli and strong ability for injury repair. Additionally, we confirmed the association between genotypes of SNPs in the strong selective gene *LCORL* and body size and muscle development in the Gushi-Anka F2 chicken population. These findings will facilitate the understanding of the Henan indigenous breeds’ germplasm traits and use potential.

## Figures and Tables

**Figure 1 animals-13-00753-f001:**
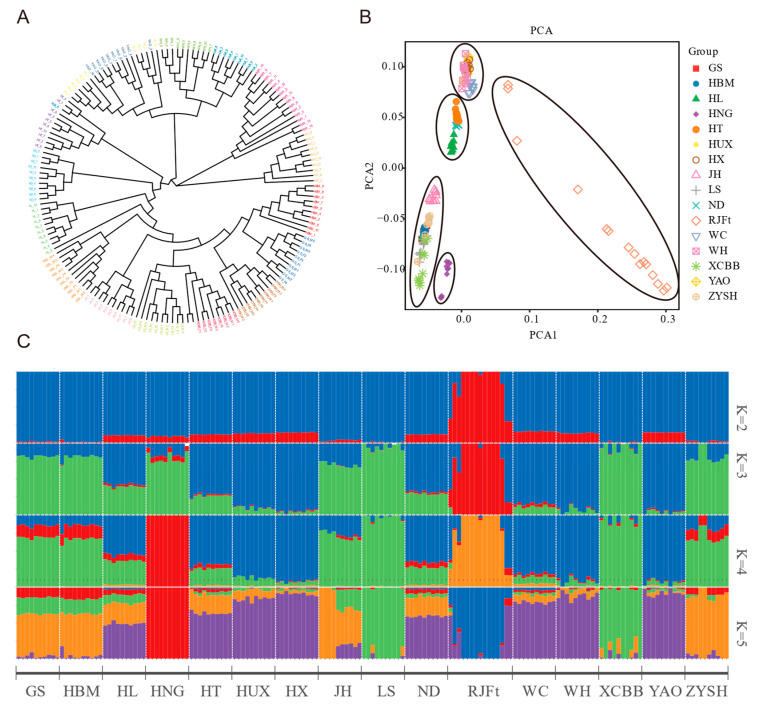
Genetic relationships and population structure between Henan indigenous chickens and other Chinese chicken breeds. (**A**) Neighbor-joining (NJ) phylogenetic tree of 16 breeds. (**B**) Principal component analysis (PCA) of 16 chicken populations. (**C**) Admixture analysis across 16 chicken populations. Proportions of genetic ancestry for 16 chicken populations with K = 2–5 (K represents the number of inferred ancestral populations. Different colors represent assumed ancestors).

**Figure 2 animals-13-00753-f002:**
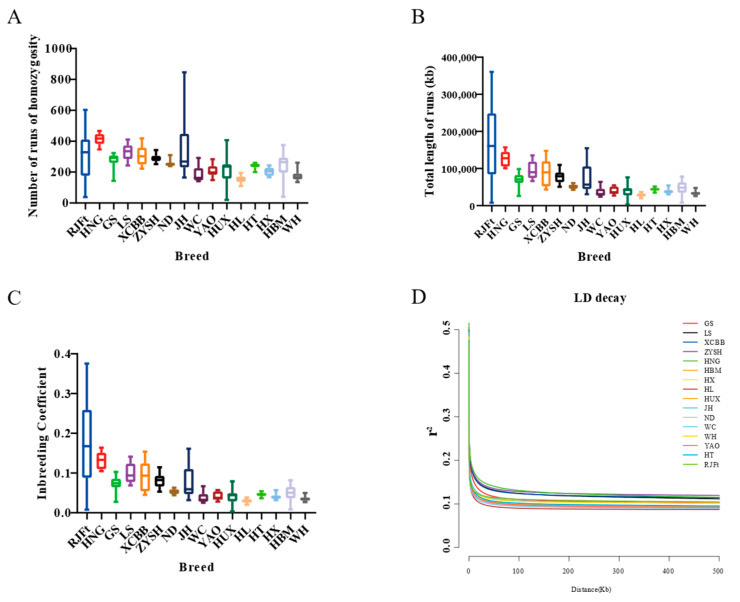
Runs of homozygosity and linkage disequilibrium of different indigenous chicken breeds and red jungle fowl. (**A**–**C**) Statistical for runs of homozygosity and genomic inbreeding coefficient of different breeds. (**D**) Decay of linkage disequilibrium in the 16 chicken populations.

**Figure 3 animals-13-00753-f003:**
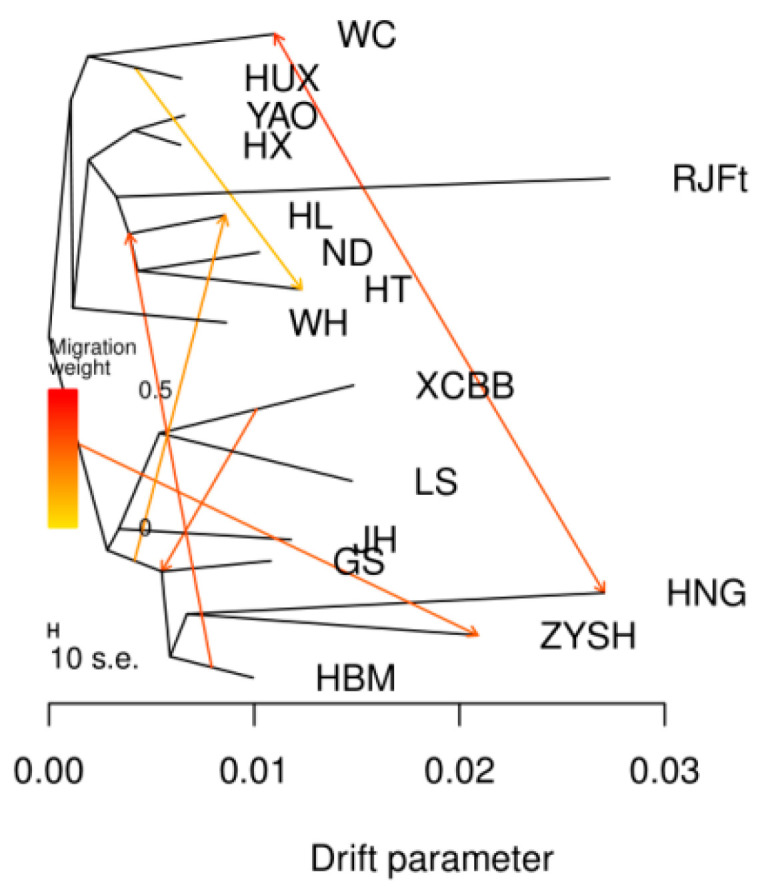
Analysis of gene flow. A maximum likelihood tree along migration events. Events relating to migration are represented as colored, weighted arrows. Parallel branch lengths are proportional to the degree of genetic drift that has appeared on each branch. The scale bar displays a standard deviation that is 10 times the mean of the values in the sample covariance matrix.

**Figure 4 animals-13-00753-f004:**
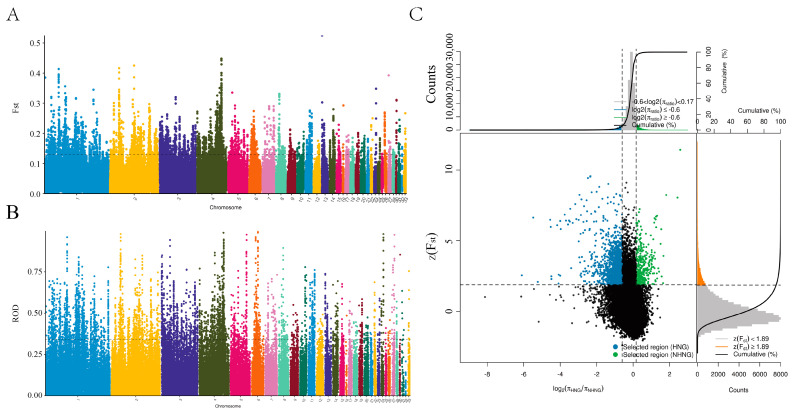
Whole genome scanning of the selection signatures between HNG and NHNG chickens. Selection signals distribution of (**A**) FST, (**B**) ROD calculated for 40 kb windows sliding in 10 kb steps. (**C**) Signals intersection by FST and ROD. Blue points represent (HNG selected genomic regions with both an extremely high Z (FST) value (top 5% level) and ROD value (top 5% level).

**Figure 5 animals-13-00753-f005:**
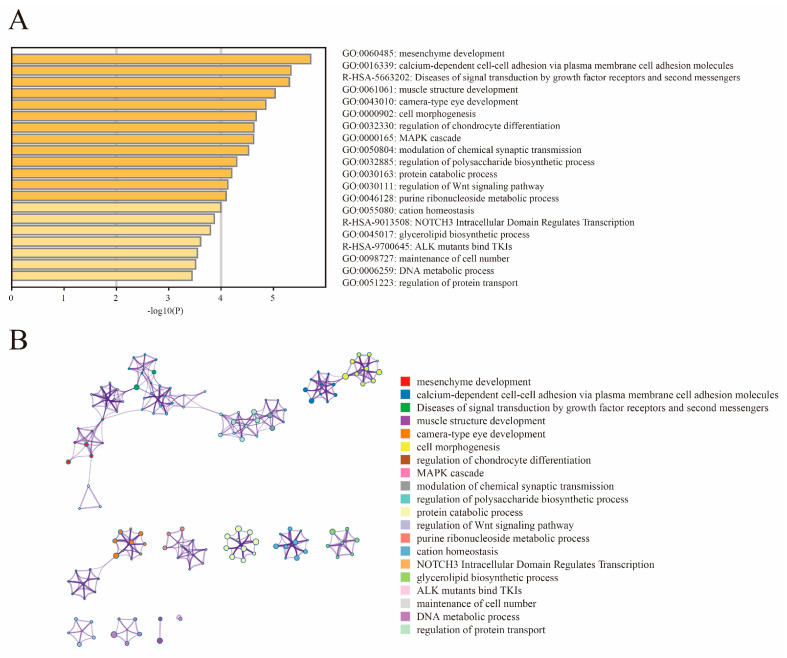
Functional enrichment analysis of 399 selected genes. (**A**) Enrichment heatmap. (**B**) Cluster analysis of enrichment pathways or biological process subclasses, colored by cluster ID, where nodes that share the same cluster ID are typically close to each other.

**Figure 6 animals-13-00753-f006:**
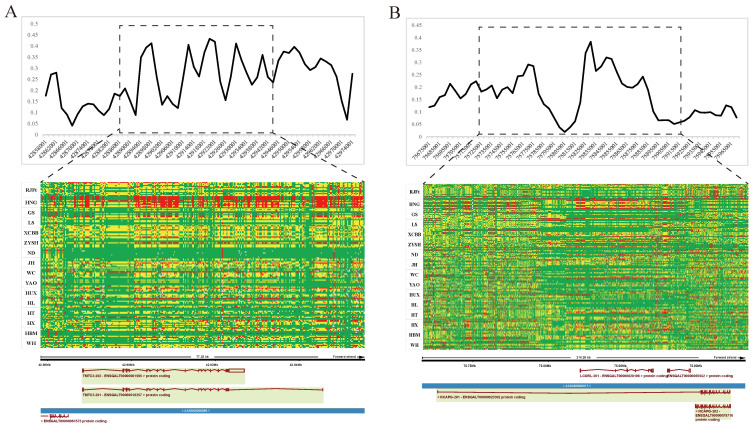
Selective sweeping signatures of *TMTC3*, *LCORL,* and *NCAPG* genes in gamecock chickens. (**A**) FST analysis and haplotype diversity of *TMTC3* genes between gamecock populations and the other non-gamecock chickens. (**B**) FST analysis and haplotype diversity of *NCAPG* and *LCORL* genes between gamecock populations and the other non-gamecock chickens. Red, yellow, and green represent homozygous mutant, heterozygous mutant, and homozygous wild-type, respectively.

**Figure 7 animals-13-00753-f007:**
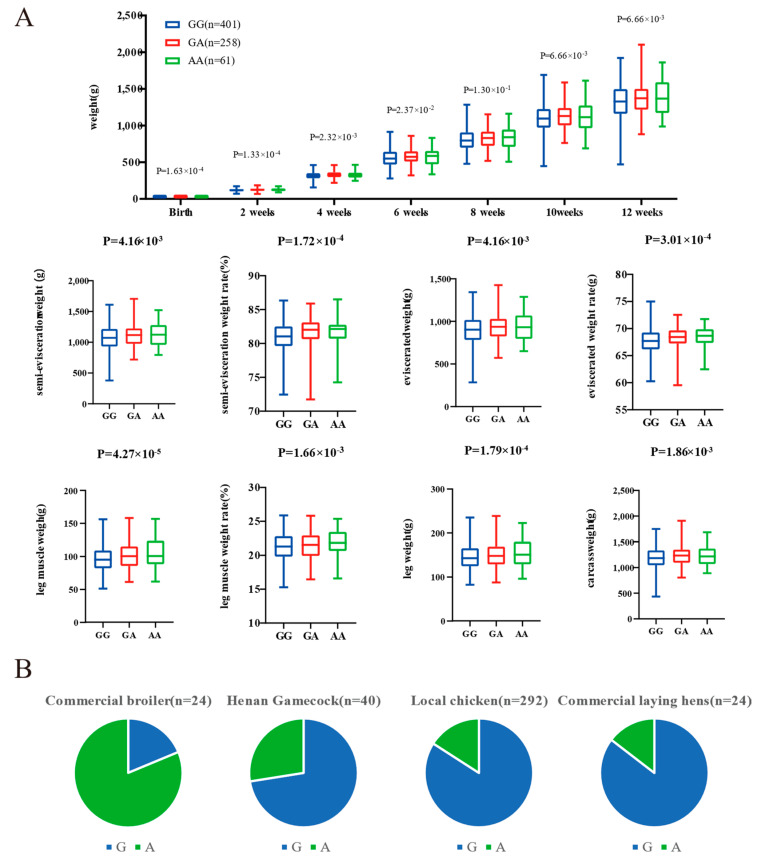
Genotype association of SNP Chr4: 75,859,000G>A in *LCORL* with body weight (BW) and carcass traits in Gushi-Anka F2 chickens (**A**), and the allelic frequency in different breeds (**B**).

**Figure 8 animals-13-00753-f008:**
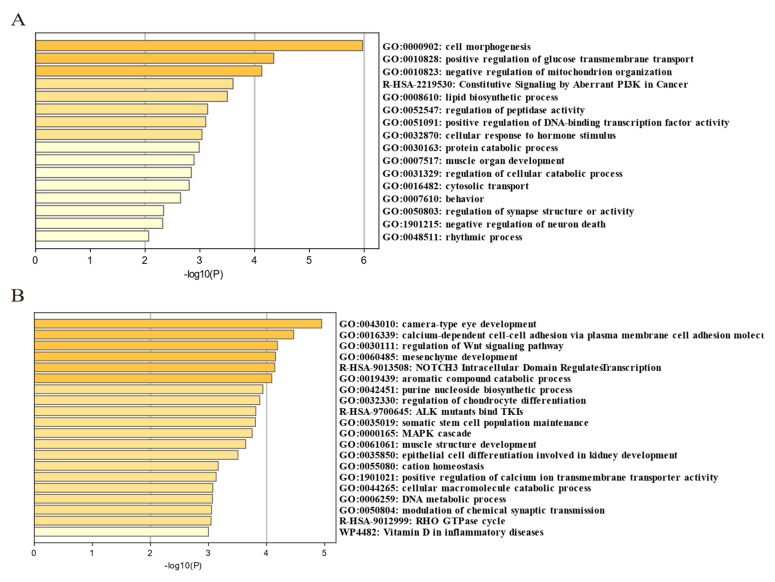
Enrichment analysis of common selected genes in gamecock chickens and those specific to HNG using Metascape. (**A**) Functional enrichment analysis of 52 common selected genes. (**B**) Functional enrichment analysis of 347 unique selected genes.

**Table 1 animals-13-00753-t001:** The genetic diversity estimates for different indigenous chicken breeds and red jungle fowl.

Population	Ho ^1^	He ^2^	π ^3^
HNG	0.288533	0.272105	0.003004
RJF	0.262515	0.278995	0.003015
JH	0.263002	0.282786	0.003061
HBM	0.281456	0.284406	0.00308
HUX	0.272044	0.289957	0.003093
WH	0.286775	0.287678	0.00312
ND	0.275747	0.288155	0.003126
LS	0.30188	0.283498	0.003133
ZYSH	0.30586	0.284161	0.003137
XCBB	0.305247	0.284792	0.003144
HT	0.285964	0.290634	0.003156
WC	0.278444	0.293583	0.003163
HX	0.283225	0.293124	0.003175
GS	0.305815	0.28887	0.003188
YAO	0.285663	0.293095	0.00319
HL	0.288032	0.297304	0.003224

^1^ Ho = Observed heterozygosity, ^2^ He = Expected heterozygosity, ^3^ π = Nucleotide diversity.

## Data Availability

All the sequence data generated in this study have been deposited in the National Genomics Data center https://bigd.big.ac.cn (accessed on 18 October 2022) with the accession codes PRJCA012571. Downloaded sequence data used in this study were presented in, online.

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
