# Peer review of "Genome-Wide Genetic Structure of Henan Indigenous Chicken Breeds"

_animals, 2023, doi:10.3390/ani13040753_

Round 1
Reviewer 1 Report
The manuscript "Genome-wide genetic structure of Henan indigenous chicken breeds and the selection signatures in Henan gamecock" is logically written and the data for the most part are convincing. Authors represent high quality original data that is clearly summarized in 8 main figures.
However the following questions should be addressed before publication:
Major comments:
C1 “Institutional Review Board Statement: The animal experiments were performed according to the Guide for the Care and Use of Laboratory Animals (Ministry of Science and Technology, China, 480 2004).” – It seems that the authors mean guide 'Guide for the Care and Use of Laboratory Animals: Eighth Edition' by National Research Council (ISBN-10 0309154006).
C2 “Data Availability Statement: All the sequence data generated in this study have been deposited in the National Genomics Data center (https://bigd.big.ac.cn) with the accession codes PRJCA012571” – The access to these data should be made open before publication.
Minor comments:
1. Title Consider removing “and the selection signatures in Henan gamecock”
2. Lines 20-22. Remove duplicated sentence
3. It would be beneficial to indicate more clearly why the analysis of “characteristics of genetic variations in the breeds” was carried out.
4. Line 192. Please, explain what the following words mean “upstreams (10.141%), and downstreams (9.625%) of genes”. How gene borders were defined?
5. Throughout the text to refer to “Henan gamecock” please, use “HNG”
6. Line 271. It is essential to describe in more detail how the results obtained can be used to draw conclusions about “migration history”.
7. Line 286 “Genome-wide selective sweep signals in Henan gamecock”. It is worth justifying why these particular stages of the study were carried out.
8. Line 305. Please indicate what was the purpose of “Cluster analysis of enrichment pathways or biological process subclasses”.
9. Please, provide explanations for the Figure 5B.
10. Conclusions. Consider removing from the conclusion section sentences with the words “might” and “suggest” that seem to be too speculative.
Consider removing this sentence, since information about the connection with these processes is given only in the discussion and with reference to other sources: “Moreover, we identified several common selective genes and biological processes/pathways in gamecocks that were related to body size, head control, muscle development, reproduction and aggression control.”
Consider removing this sentence, since the link between the identified processes and “rapid response to external stimuli” \ “injury repairing” is speculative: “Importantly, we detected the unique selective genes and biological processes related to sensory organ development and regulation of chondrocyte differentiation in Henan gamecock chickens, suggesting its rapid response to external stimuli and strong ability to injury repairing.”
Author Response
We thank you for your suggestions and comments on our manuscript. I have modified it according to your suggestion. Please see the attachment

Reviewer 2 Report
In this manuscript, the authors analyzed the genetic structure, genomic diversity and migration history of Henan indigenous chicken populations. This is an interesting study, and the findings provides deeper insights into germplasm traits and utilization potential of Henan indigenous chicken breeds. The manuscript is well written, the results are well presented, and the discussion is effective. However, some minor comments were as follows.
1. Only the full names of symbols should be provided at the first occurrence, and avoid the re-occurrence. For example, the full names of GS, ZYSH, XCBB were firstly provided in Introduction section, but were repeated in and Materials and Methods section. Please check and correct the full text.
2. Ln120: Is a redundancy the sentence“. with default parameters.”?
3. Ln147: There are two duplicated full stops in this sentence, please revise.
4. Please change [log2(ratio)] to [log2 ratio], change [log2(θπ ratio)] to [log2 (θπ ratio)], in which [2] is the subscript.
5. When using ADMIXTURE, why only 2-5 genetic clusters were shown? Which genetic cluster has the lowest cross-validation error?
6. In Figure 3, there is a covering between figure legend “Migration weight” with the branch of the ML tree. Moreover, the axis titles of Figure 2D, 4A, 4B, 4C, 5A and 5B is unclear, please optimize.
7. In my opinion, a line showing top 5% level in Figure 4 is more popular.
8. What do the red, yellow and green spots mean in Figure 6? Please explain it in the figure legends.
9. Ln 419-436: Symbols for genes should be italicized. And the full name of gene symbols should be provided at the first occurrence.
10. Compared with Chinese gamecocks, the specific candidate genome regions and genes screened in HNG may explain the formation of HNG characteristics. This manuscript aims to introduce the characteristics of germplasm resources in Henan Province, so it is suggested to add a detailed discussion on HNG-specific selective genomic regions and genes.
Author Response

(The authors gave the same response as above.)
